# Association of the PaO2/RDW ratio with 7-day mortality and risk of early invasive mechanical ventilation in ICU patients with delirium associated with ARDS: A retrospective cohort study from the MIMIC-IV database

**Jiao Xu[1], Jun Jin[2], Shan Zou[2], Si-Hao Zheng[2], Qing-Shan Zhou[2], Jiang-Tao Deng[2]\***

**1** Department of Anesthesiology, The Eighth Affiliated Hospital of Sun Yat-sen University, Shenzhen, Guangdong, China, **2** Departments of Critical Care Medicine, The University of Hong Kong-Shenzhen Hospital, Shenzhen, Guangdong, China

\* lopdeng005@gmail.com

## Abstract

### Background

Delirium is a common complication in patients with acute respiratory distress syndrome (ARDS) and is associated with poor clinical outcomes. However, studies investigating the associations between easily accessible biomarkers and early mortality or the risk of early invasive mechanical ventilation in this population remain poorly defined. This study aimed to investigate the association between the ratio of arterial partial pressure of oxygen to red cell distribution width (PaO2/RDW) and short-term outcomes in ICU patients with delirium associated with ARDS.

### Methods

Data were extracted from the Medical Information Mart for Intensive Care IV (MIMIC-IV, version 3.1), a large, publicly available critical care database that contains de-identified health records of patients admitted to Beth Israel Deaconess Medical Center. Adult ARDS patients with at least one positive Confusion Assessment Method for the ICU (CAM-ICU) evaluation were included. The primary outcome was all-cause mortality within seven days after delirium onset, and the secondary outcome was the initiation of invasive mechanical ventilation after ICU admission. Cox proportional hazards and cause-specific Cox regression models were applied to evaluate the associations between the PaO2/RDW ratio and clinical outcomes. Restricted cubic spline (RCS) modeling was used to explore potential nonlinear relationships, and subgroup analyses were performed to assess consistency across clinical strata.

**Data availability statement:** The data underlying this study are publicly available in the Medical Information Mart for Intensive Care IV (MIMIC-IV) database (version 3.1), accessible at https://physionet.org/content/mimiciv/3.1/. Access to the MIMIC-IV database requires completion of a recognized data use agreement and ethical training through the Collaborative Institutional Training Initiative (CITI) program. The authors had no special access privileges to the data, and others can access the data in the same manner.

**Funding:** The author(s) received no specific funding for this work.

**Competing interests:** The authors have declared that no competing interests exist.

## Results

A total of 4,116 patients with ARDS were initially identified, and 1,665 patients with delirium were ultimately included in the final analysis. Compared with the highest PaO2/RDW tertile, patients in the lowest tertile had significantly higher risks of 7-day (adjusted HR = 2.12, 95% CI 1.46–3.09, P < 0.001) and 30-day mortality (adjusted HR = 1.72, 95% CI 1.33–2.22, P < 0.001). The lowest tertile was also associated with an increased risk of invasive mechanical ventilation (adjusted HR = 2.68, 95% CI 1.16–6.22, P = 0.021). Restricted cubic spline analysis revealed a U-shaped association between the PaO2/RDW ratio and 7-day mortality, with the lowest estimated hazard at approximately 6.7. Subgroup analyses showed consistent associations across age, sex, and comorbidity strata without significant interactions (P for interaction > 0.05).

## Conclusions

The PaO2/RDW ratio was independently associated with 7-day mortality after delirium onset and with the early risk of invasive mechanical ventilation among ICU patients with delirium associated with ARDS. As an easily obtainable composite index, the PaO2/RDW ratio may serve as a convenient and informative biomarker for early risk assessment and clinical decision-making in critical care settings.

## Introduction

Acute respiratory distress syndrome (ARDS) is a severe form of acute lung injury characterized by refractory hypoxemia, pulmonary edema [1,2], and high mortality [3,4]. Among survivors of ARDS, delirium frequently occurs during Intensive Care Unit (ICU) stays [5], affecting up to 60–80% of mechanically ventilated patients [6], and is independently associated with increased short-term mortality [7,8], prolonged ventilation, and cognitive impairment [9,10]. The co-existence of ARDS and delirium represents a particularly high-risk clinical phenotype requiring timely identification and management with mechanical ventilation.

Currently, there is a relative lack of reliable and easily accessible biomarkers for stratifying the risks of early mortality and ventilatory management in ARDS patients complicated by delirium. The arterial partial pressure of oxygen (PaO2) reflects gas exchange efficiency [11], while red cell distribution width (RDW) is increasingly recognized as a systemic inflammation marker [12] and predictor of poor outcomes in critically ill patients [13]. A higher RDW upon admission is associated with an increased risk of long-term mortality in patients with acute respiratory failure (ARF) during a 3-year follow-up. RDW can serve as a practical and reliable prognostic marker for predicting disease progression and patient outcomes [14]. However, the clinical utility of the PaO2/RDW ratio—a composite index reflecting both oxygenation and systemic response—has not been adequately explored in this subset of patients.

Previous studies have evaluated RDW in sepsis [15] and respiratory failure [16], but its role of PaO2/RDW ratio in ARDS patients with concurrent delirium remains unclear. Arterial PaO2 serves as a direct measure of pulmonary gas exchange efficiency [17], while RDW, reflecting heterogeneity in erythrocyte size, is often elevated in states of systemic inflammation or altered erythropoiesis, conditions [18], potentially linked to hypoxia-induced tissue damage. Whether this ratio is associated with short-term mortality and early ventilation is of particular interest, given the urgent need for accessible clinical tools.

Therefore, this study aims to investigate the association between the PaO2/RDW ratio and 7-day mortality in ICU patients with delirium associated with ARDS. We also explore its relationship with early invasive mechanical ventilation following ICU admission. Our goal is to identify a simple and robust biomarker to support early risk stratification and improve clinical outcomes in this high-risk population.

## Methods

### Data access

The data utilized in this study was sourced from version 3.1 of the Medical Information Mart for Intensive Care IV (MIMIC-IV) database. This comprehensive dataset comprises de-identified patient records from individuals admitted to the emergency department or ICU at the Beth Israel Deaconess Medical Center (Boston, MA, USA) during the period from 2008 to 2022 [19]. The MIMIC-IV database was developed and maintained by the Massachusetts Institute of Technology and Beth Israel Deaconess Medical Center, where all patient data were fully de-identified in compliance with institutional and national research ethics standards. The lead author (Jiang-Tao Deng) completed the required Collaborative Institutional Training Initiative (CITI) program and obtained authorized access to the database (Certificate No. 61424658).

### Ethics statement

The study was approved by the Institutional Review Boards of the Massachusetts Institute of Technology and Beth Israel Deaconess Medical Center. The MIMIC-IV database consists of fully de-identified health data; therefore, informed consent and additional ethical approval were not required.

### Eligibility criteria

Experienced clinical researchers established the inclusion and exclusion criteria for this study.

**Inclusion criteria.** Adult patients (aged ≥18 years) were eligible for inclusion if they met the following criteria: (1) diagnosed with ARDS according to the 2023 Global Definition of ARDS; (2) admitted to the ICU for the first time during the index hospitalization; (3) had at least one PaO2 measurement and one RDW measurement obtained within the first 24 hours after ICU admission; and (4) underwent at least one documented Confusion Assessment Method for the Intensive Care Unit (CAM-ICU) evaluation during the ICU stay.

**Exclusion criteria.** The exclusion criteria were as follows: (1) patients without available CAM-ICU assessment results; (2) patients younger than 18 years; (3) patients with missing RDW or PaO2 data on the first day of ICU admission.

### Clinical variable extraction

Clinical variables were extracted from the MIMIC-IV database based on clinical relevance and data completeness. Variables with more than 5% missing values were excluded (S1 Fig). The collected information encompassed the following domains:

(1) Demographic characteristics: age and sex.

(2) Comorbidities: diabetes mellitus, hypertension, malignancy, chronic kidney disease (CKD), congestive heart failure (CHF), and chronic obstructive pulmonary disease (COPD).

(3) Laboratory parameters: baseline biochemical and hematologic indicators obtained within the first 24 hours after ICU admission, including glucose, sodium (Na), potassium (K), anion gap (AG), creatinine (Cr), blood urea nitrogen (BUN), magnesium (Mg), phosphate (Phos), total calcium (Ca), base excess (BE), partial pressure of carbon dioxide (pCO2), partial pressure of oxygen (PaO2), hemoglobin (Hb), platelet count (PLT), white blood cell count (WBC), mean corpuscular volume (MCV), red cell distribution width (RDW), and mean corpuscular hemoglobin concentration (MCHC).

(4) Physiological and clinical assessments: neurological status and disease severity within the first 24 hours after ICU admission were evaluated using the Glasgow Coma Scale (GCS) and Sequential Organ Failure Assessment (SOFA) scores, respectively.

(5) Derived index: The PaO2/RDW ratio was calculated as an integrated indicator reflecting oxygenation status relative to erythrocyte heterogeneity.

(6) Clinical outcomes included (1) all-cause mortality within seven days following the onset of delirium, and (2) the requirement for invasive mechanical ventilation after ICU admission among patients with delirium.

### Study endpoints

The primary exposure variable in this study was the PaO2/RDW ratio. The primary outcome was all-cause mortality within seven days following the onset of delirium, whereas the secondary outcome was the initiation of invasive mechanical ventilation after ICU admission among patients with delirium.

### Statistical analysis

Statistical analyses were performed using R software (version 4.4.1). The normality of continuous variables was assessed using the Shapiro–Wilk test. Non-normally distributed data were presented as medians with interquartile ranges (IQRs) and compared using the Mann–Whitney U test. Categorical variables were expressed as counts and percentages, and differences between groups were evaluated using the Chi-squared test or Fisher's exact test, as appropriate.

Kaplan–Meier (KM) survival curves were generated to evaluate the association between the PaO2/RDW ratio and 7-day mortality, with differences in survival distributions assessed using the log-rank test. Hazard ratios (HRs) and 95% confidence intervals (CIs) were estimated using Cox proportional hazards regression models. A cause-specific Cox regression model was further applied to assess the association between the PaO2/RDW ratio and the requirement for invasive mechanical ventilation after ICU admission. The proportional hazards assumption was examined using Schoenfeld residuals, and time-dependent covariates were incorporated where appropriate.

Restricted cubic spline (RCS) analysis was used to examine potential nonlinear associations of the PaO2/RDW ratio with 7-day mortality and the requirement for invasive mechanical ventilation. The overall significance was tested using the likelihood ratio test, whereas nonlinearity was evaluated using the P-nonlinear value.

Subgroup analyses were performed to assess the association between the PaO2/RDW ratio and 7-day mortality across predefined subgroups, including age, sex, comorbidities (hypertension, diabetes, CHF, malignancy, CKD, COPD), and SOFA score. Statistical significance was defined as a two-tailed $P < 0.05$.

## Results

### Patient selection and clinical characteristics

From 4,116 ARDS patients identified in the MIMIC-IV database, those younger than 18 years or without positive CAM-ICU results were excluded (n = 2,082). After further excluding patients with missing first-day data (n = 124) or missing RDW and PaO2 values (n = 245), a total of 1,665 adult patients were included in the final analysis (Fig 1). Patients were then divided into three groups according to the tertiles of the PaO2/RDW ratio.

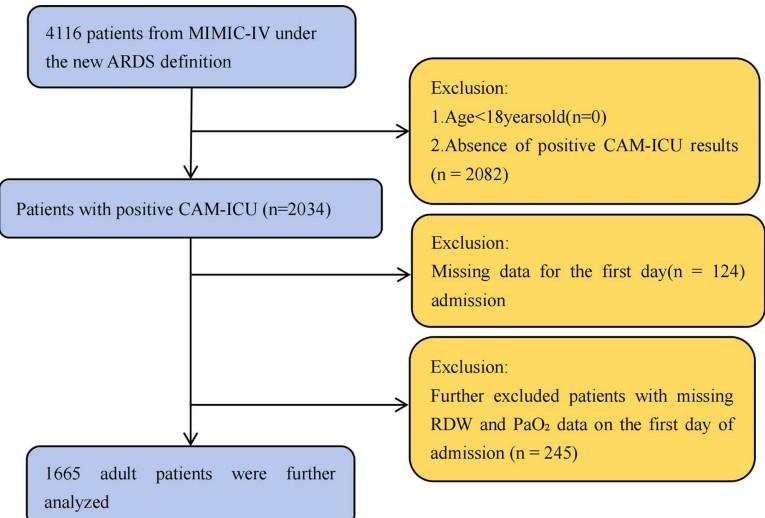

**Fig 1. Flowchart of patient selection.** Among 4,116 ICU admissions meeting the 2023 Global ARDS Definition, 2,034 had at least one positive CAM-ICU assessment. After excluding patients with missing day-1 data (n = 124) or missing RDW and PaO2 values (n = 245), 1,665 adults were included in the final analysis. Abbreviations: ARDS, acute respiratory distress syndrome; CAM-ICU, Confusion Assessment Method for the ICU; RDW, red cell distribution width; PaO2, arterial partial pressure of oxygen.

Table 1 summarizes the baseline characteristics of 1,665 ICU patients with delirium associated with ARDS, stratified by tertiles of the PaO2/RDW ratio. The normality of continuous variables was assessed using the Shapiro–Wilk test. Non-normally distributed variables were analyzed using the Kruskal–Wallis test, and categorical variables were compared using the χ² test. As shown in Table 1, age and sex distributions were similar across tertiles (P > 0.05). Significant differences were observed in several laboratory parameters. Patients in higher PaO2/RDW tertiles had higher pH, Hb, and MCHC, but lower AG, Cr, BUN, K, Mg, Phos, and BE levels (all P < 0.05). The median SOFA score decreased progressively from 6.08 in tertile 1 to 4.88 in tertile 3 (P < 0.001), indicating lower disease severity with higher ratios. Similarly, higher PaO2/RDW ratios were associated with lower rates of invasive mechanical ventilation (42.3%, 40.9%, and 35.1%; P = 0.034) and 7-day mortality (20.9%, 9.5%, and 7.6%; P < 0.001).

## Feature selection

Initially, 39 candidate variables were identified based on clinical relevance and data availability from the MIMIC-IV database. One-hot encoding was applied to unordered categorical variables. A correlation heatmap was then constructed to visualize inter-variable relationships (Fig 2). To detect multicollinearity, a correlation threshold-based approach was used, with multicollinearity defined as an absolute correlation coefficient greater than 0.9. Consequently, bicarbonate, hematocrit, PaO2, and calculated total CO2 were excluded from further analyses. Subsequently, variance inflation factor (VIF) analysis was performed to further assess multicollinearity. Variables with a VIF ≥ 5 were iteratively removed until all remaining variables had acceptable collinearity (S1 Table). To identify the most relevant variables for subsequent analyses, both least absolute shrinkage and selection operator (LASSO) regression and the Boruta algorithm were applied. As shown in Fig 3A–B, the LASSO model identified 17 variables with non-zero coefficients at the optimal regularization parameter (λ). The Boruta algorithm (Fig 3C–D) further identified 13 important variables by ranking their relative importance through random forest classification. The intersection of variables selected by both methods was considered the most stable set of predictors and was retained for subsequent multivariable regression analyses. The final selected variables included age, AG, Cr, BUN, Phos, BE, RDW, MCHC, GCS, and the PaO2/RDW ratio. In addition, the SOFA score and major comorbidities were incorporated based on clinical relevance.

**Table 1. Baseline characteristics of patients stratified by tertiles of the PaO2/RDW ratio.**

| Variables | T1 (n = 555) [2.02, 2.83] | T2 (n = 555) [3.89, 4.94] | T3 (n = 555) [6.16, 8.23] | p-value |
|---|---|---|---|---|
| Age (median [IQR]) | 70.57 [60.38, 81.16] | 68.75 [58.46, 79.26] | 70.96 [57.91, 80.42] | 0.144 |
| Female, n (%) | 301 (54.2) | 306 (55.1) | 322 (58.0) | 0.415 |
| Glucose (median [IQR]) | 131.25 [105.12, 169.00] | 134.20 [110.83, 160.15] | 136.50 [115.00, 163.30] | 0.133 |
| Na (median [IQR]) | 138.00 [134.75, 140.50] | 138.67 [136.00, 141.35] | 138.50 [136.00, 141.00] | 0.002 |
| K (median [IQR]) | 4.20 [3.81, 4.65] | 4.15 [3.85, 4.60] | 4.05 [3.78, 4.44] | 0.001 |
| pH (median [IQR]) | 7.30 [7.14, 7.37] | 7.34 [7.20, 7.39] | 7.36 [7.29, 7.42] | <0.001 |
| AG (median [IQR]) | 16.00 [13.50, 19.00] | 15.00 [12.55, 17.50] | 14.50 [12.50, 16.45] | <0.001 |
| Cr (median [IQR]) | 1.35 [0.90, 2.10] | 1.15 [0.80, 1.80] | 1.03 [0.75, 1.54] | <0.001 |
| BUN (median [IQR]) | 29.00 [19.00, 46.83] | 25.00 [16.50, 37.00] | 21.50 [14.50, 34.17] | <0.001 |
| Mg (median [IQR]) | 2.00 [1.83, 2.20] | 2.00 [1.87, 2.19] | 1.98 [1.80, 2.13] | 0.017 |
| Phos (median [IQR]) | 3.70 [2.80, 4.84] | 3.70 [2.95, 4.56] | 3.50 [2.90, 4.21] | 0.02 |
| Ca (median [IQR]) | 8.13 [7.70, 8.60] | 8.20 [7.80, 8.70] | 8.15 [7.75, 8.60] | 0.104 |
| BE (median [IQR]) | −2.00 [−5.50, 1.00] | −0.86 [−3.86, 1.67] | −0.67 [−3.40, 1.16] | <0.001 |
| pCO2 (median [IQR]) | 42.00 [35.77, 48.46] | 42.00 [36.79, 47.58] | 39.00 [35.23, 43.63] | <0.001 |
| Hb (median [IQR]) | 9.90 [8.43, 11.50] | 10.03 [8.86, 11.50] | 10.46 [9.07, 11.96] | <0.001 |
| PLT (median [IQR]) | 173.86 [117.00, 234.45] | 176.50 [127.25, 241.75] | 184.00 [134.88, 235.55] | 0.058 |
| WBC (median [IQR]) | 11.95 [8.18, 16.28] | 11.60 [8.40, 15.57] | 11.73 [8.68, 15.39] | 0.699 |
| MCV (median [IQR]) | 93.00 [88.33, 97.67] | 92.00 [87.63, 96.00] | 92.00 [88.00, 96.00] | 0.189 |
| RDW (median [IQR]) | 15.70 [14.46, 17.30] | 15.03 [13.90, 16.50] | 14.40 [13.50, 15.56] | <0.001 |
| MCHC (median [IQR]) | 32.13 [31.00, 33.17] | 32.47 [31.30, 33.51] | 32.90 [31.92, 33.82] | <0.001 |
| GCS (median [IQR]) | 14.80 [13.86, 15.00] | 15.00 [14.00, 15.00] | 15.00 [14.29, 15.00] | 0.006 |
| SOFA (median [IQR]) | 6.08 [3.96, 8.04] | 5.75 [3.88, 7.89] | 4.88 [3.08, 6.93] | <0.001 |
| PaO2/RDW (median [IQR]) | 2.41 [2.02, 2.83] | 4.47 [3.89, 4.94] | 6.93 [6.16, 8.23] | <0.001 |
| Diabetes, n (%) | 191 (34.4) | 209 (37.7) | 171 (30.8) | 0.056 |
| Hypertension, n (%) | 198 (35.7) | 245 (44.1) | 273 (49.2) | <0.001 |
| CHF, n (%) | 264 (47.6) | 224 (40.4) | 204 (36.8) | 0.001 |
| Malignancy, n (%) | 112 (20.2) | 84 (15.1) | 92 (16.6) | 0.073 |
| CKD, n (%) | 150 (27.0) | 135 (24.3) | 91 (16.4) | <0.001 |
| COPD, n (%) | 72 (13.0) | 69 (12.4) | 40 (7.2) | 0.003 |
| 7-day mortality, n (%) | 116 (20.9) | 53 (9.5) | 42 (7.6) | <0.001 |
| Invasive mechanical ventilation, n (%) | 235 (42.3) | 227 (40.9) | 195 (35.1) | 0.034 |

Abbreviations: Na, sodium; K, potassium; AG, anion gap; Cr, creatinine; BUN, blood urea nitrogen; Mg, magnesium; Phos, phosphate; Ca, calcium; BE, base excess; pCO2, partial pressure of carbon dioxide; Hb, hemoglobin; PLT, platelet count; WBC, white blood cell; MCV, mean corpuscular volume; RDW, red cell distribution width; MCHC, mean corpuscular hemoglobin concentration; GCS, Glasgow Coma Scale; SOFA, Sequential Organ Failure Assessment; PaO2/RDW, ratio of arterial partial pressure of oxygen to red cell distribution width; CHF, congestive heart failure; CKD, chronic kidney disease; COPD, chronic obstructive pulmonary disease.

### Kaplan–Meier survival curves for 7-day mortality according to PaO2/RDW tertiles

As shown in Fig 4, Kaplan–Meier analysis demonstrated a clear gradient in 7-day survival across tertiles of the PaO2/RDW ratio (overall log-rank P < 0.001). Patients in the lowest tertile (T1) had significantly lower survival compared with those in the middle (T2) and highest tertiles (T3) (T1 vs. T2, P < 0.001; T1 vs. T3, p = P < 0.001). No significant difference was observed between T2 and T3 (P > 0.05).

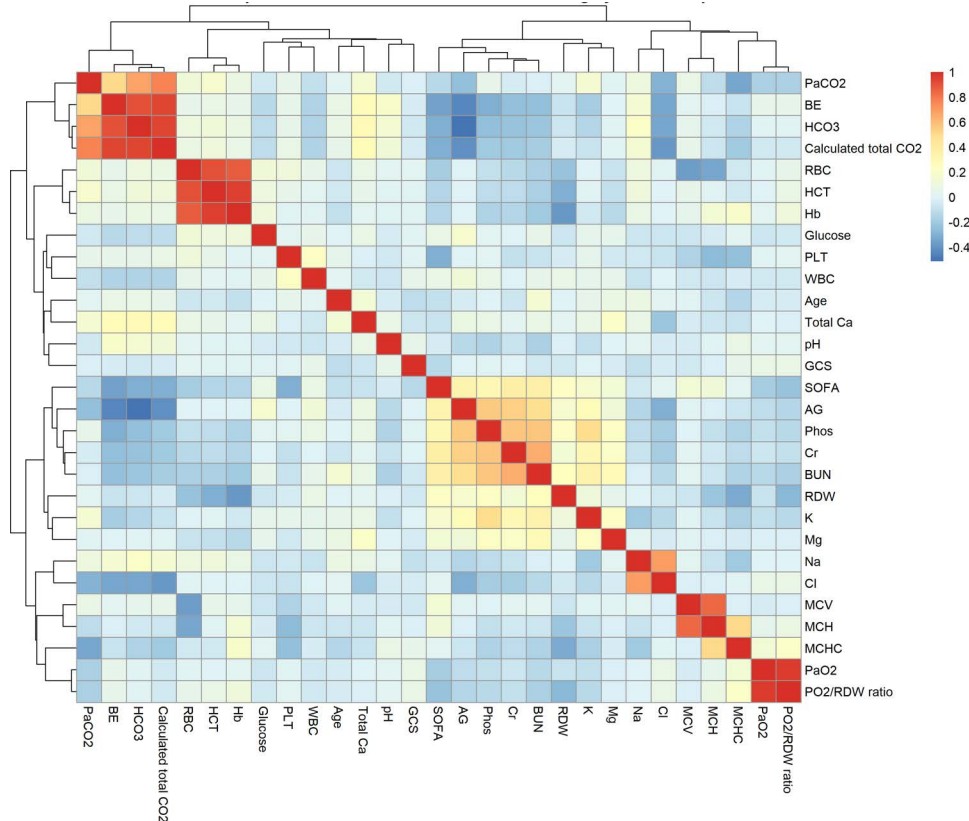

**Fig 2. Correlation heatmap of baseline variables before removal of highly correlated predictors.** The heatmap displays pairwise correlation coefficients among baseline variables. Red indicates positive correlations, and blue indicates negative correlations. Variables with an absolute correlation coefficient greater than 0.9 were considered highly correlated and subsequently excluded from further analyses. Abbreviations as in Table 1.

## Association between PaO2/RDW ratio and short-term mortality

In the Cox model for 7-day mortality following delirium, Schoenfeld residual tests, including the global test, indicated no violation of the proportional hazards (PH) assumption (all P > 0.05). In the cause-specific Cox model for time to intubation (with death treated as a competing event), both the PaO2/RDW ratio and SOFA score violated the PH assumption (P < 0.05). To address this issue, time-varying effects were modeled using interactions with log(time) via the tt() function. As shown in Fig 5, RCS analysis was conducted to evaluate the association between the PaO2/RDW ratio and 7-day mortality among ICU patients with delirium associated with ARDS. After adjustment for multiple covariates—including sex, age, AG, Cr, BUN, Phos, BE, RDW, MCHC, GCS, SOFA score within 24 hours, and comorbidities such as diabetes, hypertension, CHF, malignancy, CKD, and COPD—the RCS model revealed a significant overall association between the PaO2/RDW ratio and 7-day mortality (P-overall < 0.0001), along with a statistically significant non-linear trend (P-nonlinear = 0.0111). The HR curve displayed a U-shaped pattern, indicating variation in mortality risk across the range of PaO2/RDW ratios. The lowest estimated hazard occurred at a PaO2/RDW ratio of 6.72, which was used as the reference point. The confidence intervals widened toward both extremes of the distribution, reflecting fewer observations in these regions.

As shown in Table 2, the PaO2/RDW ratio was significantly associated with both 7-day and 30-day mortality among patients with ARDS and delirium. For 7-day mortality, each one-unit increase in the PaO2/RDW ratio was consistently associated with a lower risk of death across all models (Model 1: HR = 0.78, 95% CI 0.73–0.84; Model 2: HR = 0.80, 95% CI 0.74–0.86; Model 3: HR = 0.86, 95% CI 0.80–0.93; all P < 0.001). When stratified by tertiles, patients in the lowest

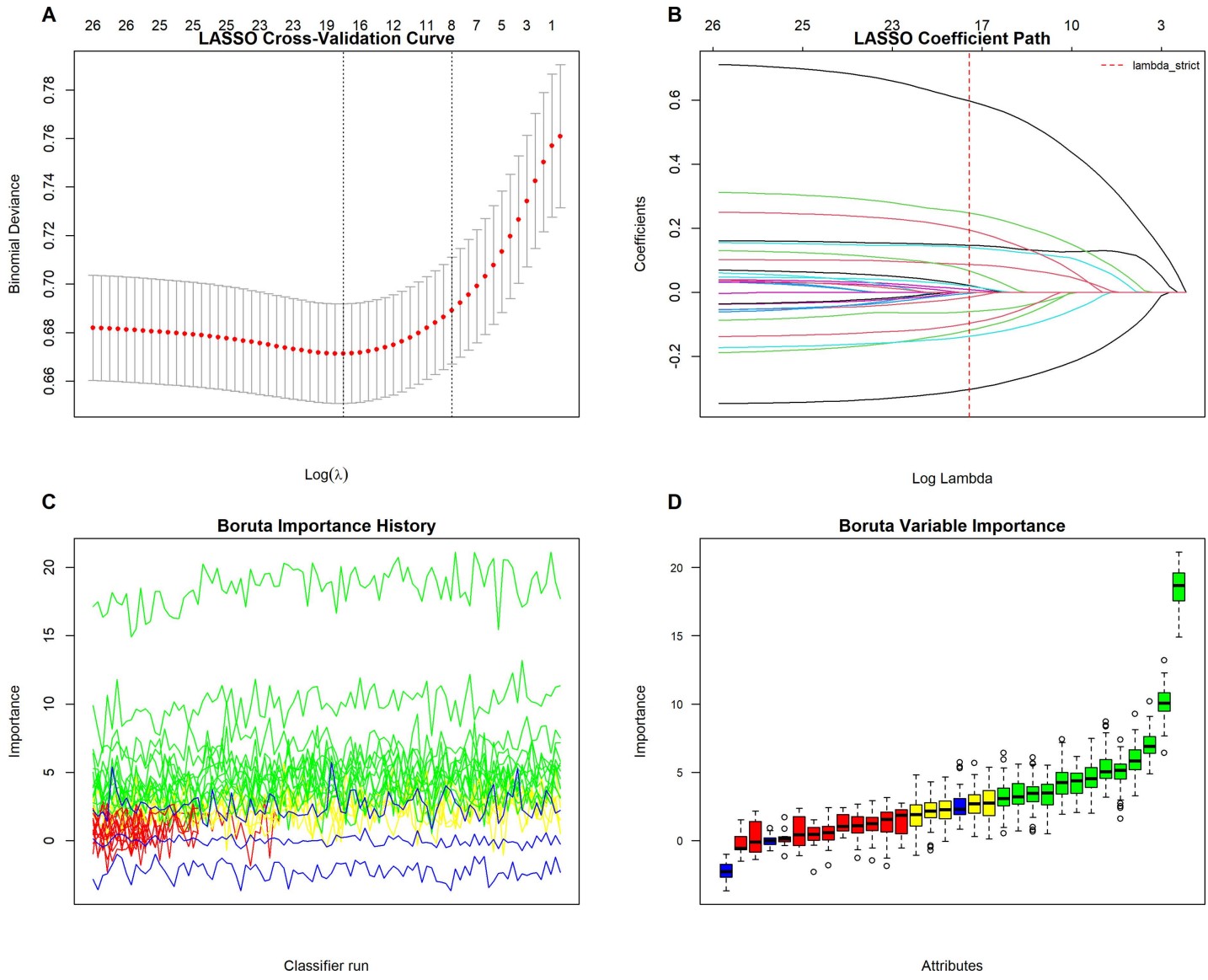

**Fig 3. Feature selection using LASSO regression and the Boruta algorithm. A.** LASSO cross-validation curve showing the mean binomial deviance versus log λ. The optimal λ (vertical dotted line) minimizes the cross-validated deviance, balancing model fit and complexity. **B.** LASSO coefficient path illustrating the shrinkage of variable coefficients toward zero as the penalty parameter λ increases. The red dashed line indicates the optimal λ selected through cross-validation. **C.** Boruta importance history across iterative classifier runs. Green lines represent confirmed important features, yellow indicate tentative features, red denote rejected features, and blue correspond to shadow attributes. **D.** Boruta variable importance plot summarizing the final importance scores. Confirmed important variables are shown in green, tentative in yellow, and rejected in red; box heights indicate variability across iterations.

tertile (T1) had a significantly higher 7-day mortality risk compared with those in the highest tertile (T3) across all models (Model 1: HR = 3.01, 95% CI 2.11–4.28; Model 2: HR = 2.90, 95% CI 2.04–4.13; Model 3: HR = 2.12, 95% CI 1.46–3.09; all P < 0.001), showing a significant linear trend (P for trend < 0.001). For 30-day mortality, similar results were observed. A one-unit increase in the PaO2/RDW ratio was associated with a reduced mortality risk (Model 1: HR = 0.85, 95% CI 0.80–0.89; Model 2: HR = 0.85, 95% CI 0.81–0.90; Model 3: HR = 0.91, 95% CI 0.86–0.96; all P < 0.001). Compared with

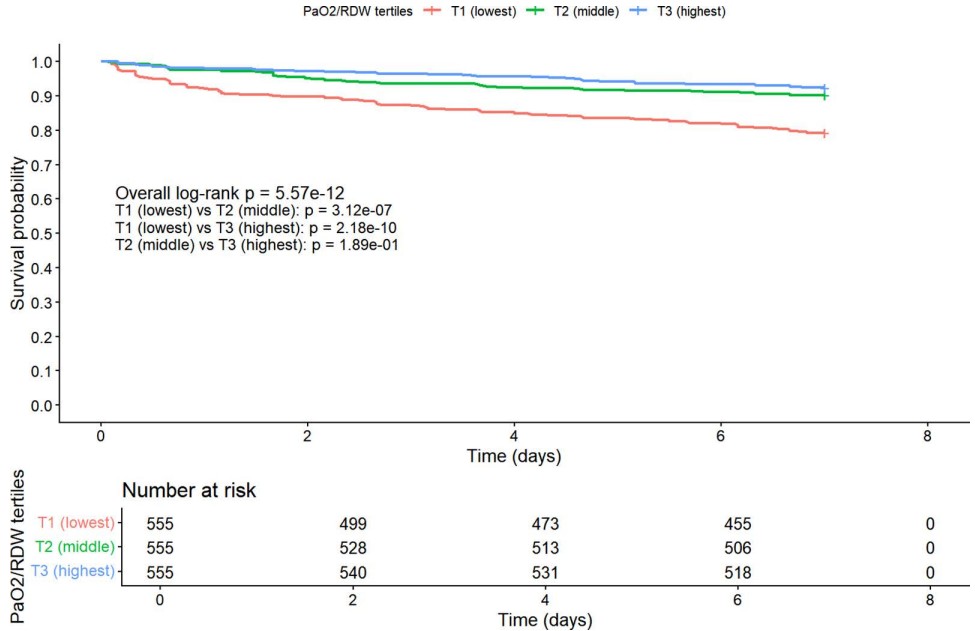

**Fig 4. Kaplan–Meier survival curves for 7-day mortality stratified by tertiles of the PaO2/RDW ratio.** Survival probability was compared among three tertile groups of the PaO2/RDW ratio (T1, lowest; T2, middle; T3, highest). Patients in the lowest tertile showed a significantly higher 7-day mortality compared with those in the middle and highest tertiles (overall log-rank P < 0.001). The number at risk for each group over time is shown below the plot. Abbreviation: PaO2/RDW, ratio of arterial partial pressure of oxygen to red cell distribution width.

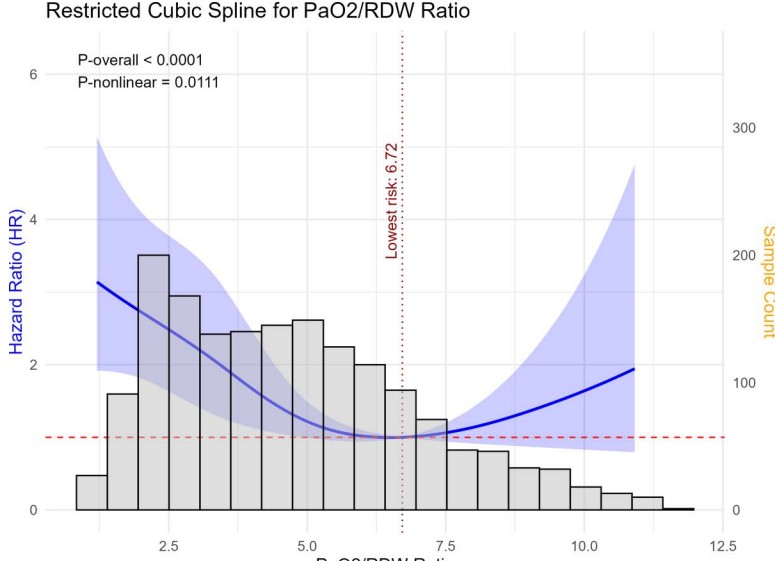

**Fig 5. Adjusted hazard ratios (blue line) and 95% confidence intervals (blue shaded area) for 7-day mortality across the continuous PaO2/RDW ratio.** The red dashed line denotes the reference level (HR = 1), and the vertical dotted line marks the point of lowest estimated risk (PaO2/RDW = 6.72). The gray histogram represents the distribution of the study population. Overall and nonlinear P-values are displayed within the panel. The model was adjusted for sex, age, AG, Cr, BUN, Phos, BE, RDW, MCHC, GCS, SOFA, diabetes, hypertension, CHF, malignancy, CKD, and COPD. Abbreviation: PaO2/RDW, ratio of arterial partial pressure of oxygen to red cell distribution width.

**Table 2. PaO2/RDW Ratio and Short-Term Mortality in ARDS With Delirium: Cox Regression Analyses for 7-Day and 30-Day Outcomes.**

| Variables | Model 1 HR (95% CI) | P-value | Model 2 HR (95% CI) | P-value | Model 3 HR (95% CI) | P-value |
|---|---|---|---|---|---|---|
| **7-day mortality** | | | | | | |
| Per 1 Unit increase | 0.78 (0.73–0.84) | < 0.001 | 0.8 (0.74–0.86) | < 0.001 | 0.86 (0.8–0.93) | < 0.001 |
| T3 (highest) | 1 (Reference) | – | 1 (Reference) | – | 1 (Reference) | – |
| T2 (middle) | 1.28 (0.85–1.92) | 0.236 | 1.33 (0.89–1.99) | 0.17 | 1.18 (0.78–1.78) | 0.424 |
| T1 (lowest) | 3.01 (2.11–4.28) | < 0.001 | 2.9 (2.04–4.13) | < 0.001 | 2.12 (1.46–3.09) | < 0.001 |
| P for trend | | < 0.001 | | < 0.001 | | < 0.001 |
| **30-day mortality** | | | | | | |
| Per 1 Unit increase | 0.85 (0.8–0.89) | < 0.001 | 0.85 (0.81–0.9) | < 0.001 | 0.91 (0.86–0.96) | < 0.001 |
| T3 (highest) | 1 (Reference) | – | 1 (Reference) | – | 1 (Reference) | – |
| T2 (middle) | 1.1 (0.84–1.45) | 0.477 | 1.14 (0.87–1.5) | 0.346 | 1.01 (0.77–1.34) | 0.919 |
| T1 (lowest) | 2.31 (1.82–2.94) | < 0.001 | 2.28 (1.79–2.9) | < 0.001 | 1.72 (1.33–2.22) | < 0.001 |
| P for trend | | < 0.001 | | < 0.001 | | < 0.001 |

Abbreviations: HR, hazard ratio; CI, confidence interval; SOFA, Sequential Organ Failure Assessment; GCS, Glasgow Coma Scale; RDW, red cell distribution width; MCHC, mean corpuscular hemoglobin concentration; BUN, blood urea nitrogen; COPD, chronic obstructive pulmonary disease; AG, anion gap; Cr, creatinine; Phos, phosphate; BE, base excess; CHF, congestive heart failure; CKD, chronic kidney disease.

Model 1: Unadjusted.

Model 2: Adjusted for age and sex.

Model 3: Adjusted for sex, age, AG, Cr, BUN, Phos, BE, RDW, MCHC, GCS, SOFA, diabetes, hypertension, CHF, malignancy, CKD, and COPD.

Tertiles: Defined according to the distribution of the PaO2/RDW ratio; the highest tertile (T3) was used as the reference category. T1 (2.02–2.80), T2 (3.91–4.92), and T3 (6.13–8.16).

the highest tertile, the lowest tertile remained significantly associated with higher 30-day mortality (Model 1: HR = 2.31, 95% CI 1.82–2.94; Model 2: HR = 2.28, 95% CI 1.79–2.90; Model 3: HR = 1.72, 95% CI 1.33–2.22; all P < 0.001), again demonstrating a significant linear trend (P for trend < 0.001).

## Association between PaO2/RDW ratio and risk of invasive mechanical ventilation

RCS analysis demonstrated a significant overall association between the PaO2/RDW ratio and the risk of invasive mechanical ventilation (P-overall = 0.0101), whereas the non-linear component was not statistically significant (P-nonlinear = 0.2291) (Fig 6). The HR curve indicated that the lowest estimated risk of invasive mechanical ventilation occurred at a PaO2/RDW ratio of 7.06. The 95% confidence intervals widened toward both extremes of the distribution, reflecting smaller sample sizes in those regions.

As shown in Table 3, the PaO2/RDW ratio was significantly associated with the risk of invasive mechanical ventilation among patients with ARDS and delirium. In the unadjusted model, each one-unit increase in the PaO2/RDW ratio was associated with a lower risk of invasive mechanical ventilation (HR = 0.70, 95% CI 0.58–0.84, P < 0.001). This association remained significant after adjustment for age and sex (Model 2: HR = 0.71, 95% CI 0.59–0.85, P < 0.001) and persisted after further adjustment for multiple covariates (Model 3: HR = 0.80, 95% CI 0.67–0.97, P = 0.022). When stratified by tertiles, patients in the lowest tertile (T1) had a significantly higher risk of invasive mechanical ventilation compared with those in the highest tertile (T3) across all models (Model 1: HR = 4.13, 95% CI 1.85–9.21; Model 2: HR = 4.18, 95% CI 1.87–9.35; Model 3: HR = 2.68, 95% CI 1.16–6.22; all P < 0.05). A significant linear trend was observed across tertiles (P for trend < 0.05).

## Subgroup analysis

As illustrated in Fig 7, the inverse association between the PaO2/RDW ratio and mortality was consistently observed across all predefined subgroups. When stratified by age, the association remained significant among patients aged

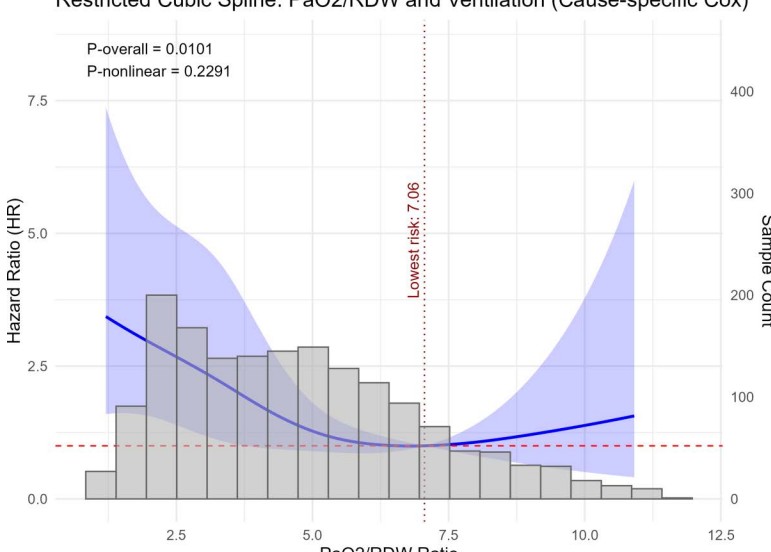

**Fig 6. Adjusted hazard ratios (blue line) and 95% confidence intervals (blue shaded area) for the risk of invasive mechanical ventilation across the continuous PaO2/RDW ratio.** The red dashed line denotes the reference level (HR = 1), and the vertical dotted line marks the point of lowest estimated risk (PaO2/RDW = 7.06). The gray histogram represents the distribution of the study sample. The overall and nonlinear P-values are displayed within the panel. The model was adjusted for sex, age, AG, Cr, BUN, Phos, BE, RDW, MCHC, GCS, SOFA, diabetes, hypertension, CHF, malignancy, CKD, and COPD. Abbreviation: PaO2/RDW, ratio of arterial partial pressure of oxygen to red cell distribution width.

**Table 3. PaO2/RDW Ratio and ventilation in ARDS With Delirium: Cause-specific Cox Regression Analysis.**

| Variables | Model 1 HR (95% CI) | P-value | Model 2 HR (95% CI) | P-value | Model 3 HR (95% CI) | P-value |
|---|---|---|---|---|---|---|
| 7-day mortality | | | | | | |
| Per 1 Unit increase | 0.7 (0.58–0.84) | < 0.001 | 0.71 (0.59–0.85) | < 0.001 | 0.8 (0.67–0.97) | 0.0221 |
| T3 (highest) | 1 (Reference) | | 1 (Reference) | | 1 (Reference) | |
| T2 (middle) | 1.73 (0.83–3.61) | 0.142 | 1.83 (0.87–3.82) | 0.109 | 1.49 (0.7–3.17) | 0.299 |
| T1 (lowest) | 4.13 (1.85–9.21) | < 0.001 | 4.18 (1.87–9.35) | < 0.001 | 2.68 (1.16–6.22) | 0.0213 |
| P for trend | | < 0.001 | | < 0.001 | | 0.0122 |

Abbreviations: HR, hazard ratio; CI, confidence interval; SOFA, Sequential Organ Failure Assessment; GCS, Glasgow Coma Scale; RDW, red cell distribution width; MCHC, mean corpuscular hemoglobin concentration; BUN, blood urea nitrogen; COPD, chronic obstructive pulmonary disease; AG, anion gap; Cr, creatinine; Phos, phosphate; BE, base excess; CHF, congestive heart failure; CKD, chronic kidney disease.

Model 1: Unadjusted.

Model 2: Adjusted for age and sex.

Model 3: Adjusted for sex, age, AG, Cr, BUN, Phos, BE, RDW, MCHC, GCS, SOFA, diabetes, hypertension, CHF, malignancy, CKD, and COPD.

Tertiles: Defined according to the distribution of the PaO2/RDW ratio; the highest tertile (T3) was used as the reference category. T1 (2.02–2.80), T2 (3.91–4.92), and T3 (6.13–8.16).

>65 years (HR = 0.76, 95% CI 0.70–0.83, P < 0.001) but was not statistically significant among those aged ≤65 years (HR = 0.89, 95% CI 0.76–1.04, P = 0.148; P for interaction = 0.079). Similar patterns were observed across sex and comorbidity subgroups, including hypertension, diabetes, heart failure, malignancy, CKD, and COPD, with all interaction P-values > 0.05. The association also remained consistent across disease severity categories stratified by SOFA score (≤5 vs. > 5; P for interaction = 0.128). Collectively, no statistically significant effect modification was identified in any of the examined subgroups.

## Subgroup Analysis of PaO2/RDW and Mortality

| Subgroup | Death (n%) | HR (95% CI) | P-value | P for interaction |
|---|---|---|---|---|
| **Sex** | | | | 0.196 |
| Male | 95 (12.9%) | 0.83 (0.74-0.92) | <0.001 | |
| Female | 116 (12.5%) | 0.75 (0.67-0.83) | <0.001 | |
| **Age** | | | | 0.079 |
| >65 | 173 (16.7%) | 0.76 (0.70-0.83) | <0.001 | |
| ≤65 | 38 (6%) | 0.89 (0.76-1.04) | 0.148 | |
| **SOFA** | | | | 0.128 |
| ≤5 | 72 (9.9%) | 0.85 (0.76-0.95) | 0.003 | |
| >5 | 139 (14.9%) | 0.75 (0.68-0.83) | <0.001 | |
| **Hypertension** | | | | 0.085 |
| Yes | 92 (12.8%) | 0.84 (0.75-0.93) | <0.001 | |
| No | 119 (12.5%) | 0.73 (0.66-0.82) | <0.001 | |
| **Diabetes** | | | | 0.085 |
| No | 146 (13.3%) | 0.81 (0.75-0.89) | <0.001 | |
| Yes | 65 (11.4%) | 0.70 (0.60-0.81) | <0.001 | |
| **Heartfailure** | | | | 0.921 |
| No | 104 (10.7%) | 0.79 (0.71-0.87) | <0.001 | |
| Yes | 107 (15.5%) | 0.79 (0.71-0.88) | <0.001 | |
| **Malignancy** | | | | 0.143 |
| No | 162 (11.8%) | 0.76 (0.70-0.83) | <0.001 | |
| Yes | 49 (17%) | 0.86 (0.75-0.99) | 0.034 | |
| **CKD** | | | | 0.378 |
| No | 150 (11.6%) | 0.80 (0.74-0.87) | <0.001 | |
| Yes | 61 (16.2%) | 0.74 (0.63-0.87) | <0.001 | |
| **COPD** | | | | 0.136 |
| Yes | 23 (12.7%) | 0.92 (0.74-1.14) | 0.442 | |
| No | 188 (12.7%) | 0.77 (0.71-0.83) | <0.001 | |

Hazard Ratio (HR): 0.50 — 0.71 — 1.0

**Fig 7. Forest plot showing adjusted hazard ratios (HRs) and 95% confidence intervals across clinical subgroups.** No significant interactions were observed between the PaO2/RDW ratio and any subgroup variables (all P for interaction > 0.05). Abbreviations: PaO2, partial pressure of oxygen; RDW, red cell distribution width; CKD, chronic kidney disease; COPD, chronic obstructive pulmonary disease; Cr, creatinine; AG, anion gap; BUN, blood urea nitrogen; Phos, phosphate; BE, base excess; MCHC, mean corpuscular hemoglobin concentration; SOFA, Sequential Organ Failure Assessment; GCS, Glasgow Coma Scale.

## Discussion

In this large retrospective cohort of critically ill patients with ARDS complicated by delirium, we observed a significant association between the PaO2/RDW ratio and short-term clinical outcomes. Lower PaO2/RDW ratios were correlated with higher 7-day and 30-day mortality rates following delirium onset, as well as an increased likelihood of requiring invasive mechanical ventilation after ICU admission. Restricted cubic spline analysis revealed a U-shaped association between the PaO2/RDW ratio and mortality, with the lowest estimated risk observed at approximately 6.7. Subgroup analyses demonstrated that these associations remained consistent across different demographic and clinical subgroups, without significant effect modification. Collectively, these findings indicate that the PaO2/RDW ratio—an integrated indicator reflecting both oxygenation efficiency and erythrocyte heterogeneity—is closely associated with disease severity and may serve as a convenient and informative marker for early risk assessment in ICU patients with delirium associated with ARDS.

The observed relationship between the PaO2/RDW ratio and adverse outcomes may reflect the combined impact of impaired oxygenation and systemic inflammation. PaO2 serves as a direct indicator of pulmonary gas exchange efficiency [20], whereas RDW reflects red blood cell size heterogeneity and has been linked to oxidative stress, inflammation, and multiorgan dysfunction [21–23] in critically ill patients. A lower PaO2/RDW ratio therefore indicates concurrent hypoxemia and heightened inflammatory burden, both of which are key determinants of poor prognosis in ARDS. Previous studies have reported that elevated RDW is associated with increased mortality in sepsis [24], respiratory failure [25], and critical illness [26], consistent with our findings. The composite index integrating PaO2 and RDW may offer a more comprehensive value by capturing both pulmonary oxygenation impairment and systemic inflammatory or metabolic disturbances. This dual-dimensional assessment could not only explain its strong association with disease severity but also with the need for mechanical ventilation.

Mechanistically, several factors may underlie this association. Hypoxia stimulates erythropoietin release [27] and disrupts erythropoiesis [28], resulting in a greater proportion of immature reticulocytes and elevated RDW [29]. Moreover,

inflammatory cytokines and oxidative stress may impair red blood cell membrane stability and shorten erythrocyte lifespan, further increasing size variability [30,31]. Finally, hypoxia-induced inflammation can impair red blood cell maturation, causing a heterogeneous population of red blood cells and an increased RDW [32]. These processes collectively contribute to tissue hypoxia and organ dysfunction, such as increased susceptibility to brain dysfunction, leading to delirium and thereby explaining the higher mortality observed in patients with lower PaO2/RDW ratios. The U-shaped pattern observed in our spline analysis suggests that both inadequate and excessive oxygenation may be detrimental—a finding that aligns with prior evidence that hyperoxia can induce oxidative injury and worsen outcomes in ARDS [33,34].

Importantly, our findings extend beyond mortality, highlighting that the composite index is also associated with the initiation of invasive mechanical ventilation after ICU admission in patients with delirium. A lower PaO2/RDW ratio was associated with an increased need for early invasive mechanical ventilation, implying that this index may help identify patients at risk of respiratory deterioration. Given that RDW reflects systemic stress and PaO2 indicates oxygenation efficiency [35–37], their ratio may represent an integrated marker of respiratory resilience and overall physiological reserve. PaO2 alone overlooks other critical dimensions of respiratory physiology, including carbon dioxide clearance and respiratory muscle performance [38]. Integrating the PaO2/RDW ratio into routine assessment may thus support early identification of decompensation, guide individualized ventilatory management, and ultimately enhance patient outcomes.

Clinically, the PaO2/RDW ratio offers several advantages. Both parameters are routinely measured, inexpensive, and readily available [39,40] in the ICU setting. Compared with complex scoring systems, this ratio provides a simple yet physiologically meaningful tool for early risk stratification. If validated prospectively, it could complement established severity indices such as the SOFA score to guide treatment intensity and monitoring frequency in ICU patients with delirium associated with ARDS.

Nevertheless, several limitations should be acknowledged. First, this was a retrospective study based on the MIMIC-IV database, which may introduce potential selection bias and residual confounding. Although multiple covariates were adjusted for, unmeasured factors such as ventilator settings, sedation practices, and transfusion history might still have influenced the results. Second, the MIMIC-IV database represents a single-center cohort (Beth Israel Deaconess Medical Center), which may limit the generalizability of our findings to other institutions or healthcare systems. Third, variations in the timing of measurements and interventions could affect the accuracy of exposure and outcome assessments. Fourth, the observational nature of this study precludes causal inference, and the observed associations should be interpreted as hypothesis-generating rather than definitive. Finally, changes in ARDS management and critical care practices over the study period (2008–2022) may also have influenced patient outcomes. Future multicenter, prospective studies and external validations are warranted to confirm these findings and to elucidate the underlying biological mechanisms linking the PaO2/RDW ratio to outcomes in ARDS-associated delirium.

In summary, this study demonstrates that a lower PaO2/RDW ratio is independently associated with increased short-term mortality and a higher risk of early invasive mechanical ventilation among ICU patients with delirium associated with ARDS. As a simple and easily obtainable index that reflects both oxygenation and systemic response, the PaO2/RDW ratio may serve as a valuable adjunct for early risk assessment and clinical decision-making in critical care.

## Conclusion

In this retrospective cohort study of critically ill patients with ARDS complicated by delirium, a lower PaO2/RDW ratio was independently associated with higher short-term mortality and an increased risk of early invasive mechanical ventilation. These findings suggest that the PaO2/RDW ratio—an easily obtainable and physiologically integrated index reflecting both oxygenation efficiency and systemic response—may serve as a practical biomarker for early risk stratification and clinical management in ICU patients with delirium associated with ARDS.

## Supporting information

**S1 Table. Multicollinearity analysis and variance inflation factor (VIF) screening.** This table displays the iterative process of removing variables with high VIF values to reduce multicollinearity. Columns include the iteration number, variable name, and the calculated VIF value.
(CSV)

**S1 Fig. Percentage of missing values for candidate variables.** This figure illustrates the proportion of missing data for each clinical variable in the initial dataset. Variables with missing values exceeding 5% were excluded from the final analysis.
(TIF)

## Acknowledgments

The authors gratefully acknowledge the Massachusetts Institute of Technology and Beth Israel Deaconess Medical Center for maintaining the MIMIC-IV database and granting open access for research use. We also appreciate the efforts of all contributors to the PhysioNet platform for supporting data sharing and open science.

## Author contributions

**Conceptualization:** Jun Jin, Qing-Shan Zhou, Jiang-Tao Deng.

**Data curation:** Jiao Xu.

**Methodology:** Shan Zou, Si-Hao Zheng.

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
