## [Decision Letter · Decision Letter 0]

6 Oct 2025

Dear Dr. deng,

Thank you for submitting your manuscript to PLOS ONE. After careful consideration, we feel that it has merit but does not fully meet PLOS ONE’s publication criteria as it currently stands. Therefore, we invite you to submit a revised version of the manuscript that addresses the points raised during the review process.

**ACADEMIC EDITOR: Major revision**

We look forward to receiving your revised manuscript.

Kind regards,

Marwan Salih Al-Nimer, MD, PhD

Academic Editor

PLOS ONE

Journal Requirements:

2. For studies involving third-party data, we encourage authors to share any data specific to their analyses that they can legally distribute. PLOS recognizes, however, that authors may be using third-party data they do not have the rights to share. When third-party data cannot be publicly shared, authors must provide all information necessary for interested researchers to apply to gain access to the data. (https://journals.plos.org/plosone/s/data-availability#loc-acceptable-data-access-restrictions)

4. Please amend the manuscript submission data (via Edit Submission) to include author Qing-Shan Zhou

5. Please amend your authorship list in your manuscript file to include author lop – deng

Additional Editor Comments:

Kindly adhere to the guidelines of the journal (references)

Reviewers' comments:

Reviewer's Responses to Questions

**Comments to the Author**

1. Is the manuscript technically sound, and do the data support the conclusions?

Reviewer #1: Yes

Reviewer #2: Partly

2. Has the statistical analysis been performed appropriately and rigorously?

Reviewer #1: Yes

Reviewer #2: No

3. Have the authors made all data underlying the findings in their manuscript fully available?

Reviewer #1: Yes

Reviewer #2: Yes

4. Is the manuscript presented in an intelligible fashion and written in standard English?

Reviewer #1: Yes

Reviewer #2: Yes

Reviewer #1: Some vocabulary choices and phrasing throughout the manuscript could be revised to enhance clarity and improve readability. Refining the language and writing style would help make the content more accessible and better illustrated for the reader.

The manuscript would benefit from further improvements, as outlined in my comments to the authors below.

1. In the abstract, line 3, it is better to use ‘’Poorly defined’’ instead of ‘’lacking’’.

2. In the abstract, line 5, I recommend using “delirium associated with ARDS in ICU patients” instead of “ARDS patients with ICU-acquired delirium.” The current phrasing may be misleading, as it suggests the delirium is primarily associated with ICU admission rather than with ARDS itself. Based on the manuscript, it appears that the authors intend to highlight delirium in the context of ARDS within the ICU setting. To avoid confusion and better reflect the study’s focus, the authors should revise this wording accordingly.

3. While '’ICU'’ is widely recognized by clinicians, defining the term in the abstract is recommended to ensure clarity for all readers.

4. In the abstract, a definition of the MIMIC-IV database should be provided.

5. In the abstract, it is recommended to mention that a total of 4,116 patients were initially identified from the MIMIC-IV database, rather than only stating that the final cohort consisted of 1,508 patients. Including this information provides clearer context on the study population and the selection process.

6. In the abstract, conclusions, it's preferable to state that the PO2/RDW ratio may offer predictive insights, rather than claiming it is a reliable biomarker for predicting 7-day mortality and guiding mechanical ventilation in ARDS patients with delirium.

7. As previously mentioned, the study aims in the Introduction should be revised for clarity. I recommend using the phrase “delirium associated with ARDS in ICU patients” instead of “ARDS patients with ICU-acquired delirium.” The current wording may be misleading, as it implies that delirium is primarily linked to ICU admission rather than ARDS. Based on the manuscript, it seems the authors aim to emphasize delirium in the context of ARDS within the ICU setting. To better reflect this focus and avoid misinterpretation, the authors should adjust the phrasing accordingly.

8. There is still room to improve the Introduction. I recommend revising it to provide a more comprehensive overview of the topic, including citing additional relevant studies. Strengthening the background will help readers better understand the context and significance of the study, as well as clarify its specific aims. These enhancements can be effectively addressed within the Introduction.

9. The Methods section should be presented more clearly and in greater detail to help readers fully understand the study design, analysis process, and related procedures. For example, the statement “Eligibility criteria: Experienced clinical researchers defined the inclusion and exclusion criteria for this study” is too vague. Although some related points are included, there is still a good opportunity to further clarify and expand on the selection criteria. This would improve the overall clarity and quality of the manuscript.

10. Although the authors have made an effort to present the results in a way that is understandable, the Results section would benefit from clearer and more structured writing, as it can be difficult at times for the reader to follow the flow of information. One key issue is the lack of detailed figure legends. Currently, only figure titles are provided, without sufficient explanation regarding the methods used, definitions applied, or interpretation of results. Including comprehensive figure legends is essential and would greatly aid reader comprehension. Overall, there is room for improvement in how the results are presented to ensure they are more straightforward and easier to follow.

11. The Discussion would benefit from revision to provide a more comprehensive overview of the relevance of the findings in the context of existing literature. It is recommended to improve the interpretation of the results by clearly linking them to prior research. Where possible, citing additional relevant studies could strengthen the discussion and enhance the overall impact of the manuscript.

12. I recommend that the Conclusion be rewritten to improve clarity and conciseness, which will enhance readability and more effectively convey the key messages to the reader.

Reviewer #2: This retrospective study of 1,508 ARDS patients with ICU-acquired delirium from MIMIC-IV explores the PO₂/RDW ratio (first 24-hour arterial PO₂ divided by RDW) as a predictor of 7-day all-cause mortality and as an indicator for mechanical ventilation. Higher PO₂/RDW was associated with lower 7-day mortality (linear trend on restricted cubic splines) and with lower odds of requiring mechanical ventilation (logistic model). The authors use Kaplan–Meier, Cox regression (with covariate selection using Boruta and SHAP), restricted cubic splines, and multivariable logistic regression.

A. Overall appraisal

Strengths

Novel, clinically intuitive composite marker combining oxygenation and a widely available hematologic parameter.

Large, well-known database (MIMIC-IV) and a plausible clinical question with potential bedside utility.

Multiple complementary analyses (KM, Cox, RCS, logistic regression, SHAP/Boruta) demonstrate depth.

However, there are important methodological, analytical, and interpretative issues that must be addressed before this is suitable for publication. Many relate to temporality/causality (timing of PO₂ vs ventilation), model specification (PH assumption, overfitting), handling of missing data, inconsistent reporting, and interpretation of effect measures. Below I list these as major and minor points with suggested remedies.

B. Major concerns

1. Inconsistent, and potentially incorrect, interpretation of hazard ratios in the Abstract / Results.

Example: the Abstract states “higher PO₂/RDW ratios were significantly associated with lower 7-day mortality, with patients in the highest quartile having a significantly reduced risk compared to those in the lowest quartile (HR: 1.90, 95% CI: 1.12–3.22; p = 0.017).” An HR of 1.90 implies higher risk in that comparison, not reduced risk. Later you report Model 3 HR for T1 = 1.90 (T4 reference) — i.e., lowest quartile has higher hazard than highest quartile.

Fix: Carefully check/reference each HR and explicitly state which group is reference. Correct wording throughout to remove this and any similar contradictions.

2. Ambiguity about timing of exposure and outcome (reverse causation / temporality).

You state PO₂/RDW was measured “within the first 24 hours” and ventilator use was assessed “after PO₂.” It is crucial to show that the PaO₂ and RDW values used as predictors were measured before the decision to intubate / initiate mechanical ventilation. If many arterial gases were obtained after ventilation or after major resuscitation, the association may reflect treatment effects, not prognosis.

Fix / analyses required:

i. Explicitly report the exact timestamp windows used for PO₂ and RDW measurement relative to ICU admission and to initiation of ventilation (median time difference and IQR).

ii. Restrict the main analysis to patients where PO₂/RDW is observed before intubation/ventilation; present a sensitivity analysis excluding measurements taken after ventilation initiation.

iii. If timing cannot be disentangled for a substantial portion, temper causal claims about ventilator guidance.

3. Confounding by indication and inappropriate adjustment for mediators.

Mechanical ventilation is likely part of the causal pathway between severity and mortality. Adjusting for ventilation when modeling mortality can induce bias (collider/mediator bias). Similarly, including variables that change after the exposure (e.g., interventions) can distort associations.

Fix / analyses required:

i. Re-run primary mortality models without adjusting for variables that plausibly lie on the causal path (e.g., ventilation), or present models both with and without ventilation and explain rationale.

ii. Consider causal diagrams (DAG) to demonstrate chosen covariates.

iii. For the mechanical ventilation outcome, use methods to control confounding (propensity score methods, inverse probability weighting) or clearly acknowledge limitations.

4. Proportional hazards (PH) assumption handling is inadequate.

You report Schoenfeld residuals showing PH violations for several covariates and state you “removed the covariates that did not meet the criteria.” Dropping covariates post-hoc because they violate PH is not an appropriate strategy and can bias results.

Fix / analyses required:

i. Use accepted methods: include time-dependent covariates for violating predictors, stratified Cox models, or alternative modelling (AFT model).

ii. Report Schoenfeld residual plots and give details of any time-dependent terms.

iii. If covariates were removed, justify and present sensitivity analyses with corrected models.

5. Events-per-variable (EPV) and risk of overfitting.

With ~1,508 patients and ~9.22% 7-day mortality (~139 events), the maximum reliable number of covariates in a Cox model is limited (~10–14 by old rule of thumb). Model 3 appears to include a long list of covariates (possibly exceeding EPV recommendations).

Fix / analyses required:

i. Report number of deaths and compute EPV.

ii. Consider penalized regression (ridge/LASSO) or reduce covariates via pre-specification (clinical importance) or dimension reduction.

iii. Provide internal validation with bootstrap or cross-validation and report optimism-corrected performance (c-index) and calibration.

6. Missing data handling is unclear.

You excluded patients with missing RDW or PO₂ but did not report how missingness for other covariates was handled (complete case? imputation?). Excluding those with other missing covariates can bias results if missing not at random.

Fix / analyses required:

i. Provide a missingness table (n and % missing per variable).

ii. If any imputation was done, describe method (multiple imputation, number of imputations, model used). If complete case analysis, justify and provide sensitivity analysis using multiple imputation.

7. Cutoff selection and clinical thresholds (Youden index) - risk of overfitting and lack of validation.

You present two different cutoffs (3.00 for mortality ref point, 4.22 for ventilation) chosen by Youden’s index. Data-driven cutpoints tend to be optimistic and need validation.

Fix / analyses required:

i. Report ROC curves, AUC (with 95% CI), sensitivity, specificity, PPV, NPV for chosen cutpoints.

ii. Use bootstrapping to estimate 95% CIs for cutpoints and performance metrics.

iii. Present decision-curve analysis to assess clinical utility. Emphasize that cutoffs require external validation.

8. Modelling of mechanical ventilation as a binary logistic outcome may be inappropriate.

Ventilation is a time-dependent event and may be censored by death; treating it as a simple binary outcome ignores timing and competing risks. Patients who die early may never be ventilated (competing risk).

Fix / analyses required:

i. Consider time-to-intubation (cause-specific hazard) using Cox models, or competing-risks methods (Fine-Gray) with death as a competing event.

ii. If keeping logistic model, carefully justify the time window (e.g., ventilation within 24–48h) and show timing distribution.

9. Feature selection and circularity concerns.

You removed PO₂ and RDW as individual features for multicollinearity, yet used PO₂/RDW as a predictor. Explain how and why these removals were done, and show correlation matrix and variance inflation factors (VIF).

Fix: Provide rationale; present models that include the ratio and also separate models using PaO₂ and RDW individually for comparison.

10. External validity, calibration, and clinical implementation not demonstrated.

A predictive marker needs discrimination and calibration assessment; you report HRs and ORs but not discrimination (c-index/AUC) or calibration plots. Also no external validation.

Fix / analyses required:

i. Provide discrimination metrics (c-index for Cox, AUC for logistic) with optimism correction.

ii. Provide calibration plots (observed vs predicted at 7 days).

iii. State plans/limitations regarding external validation; if possible, perform temporal or hospital-level split internal validation, or bootstrap validation.

C. Minor and editorial concerns

1. Terminology and consistency: Use PaO₂ or PO₂ consistently (PaO₂ is standard). Define RDW units (RDW-CV vs RDW-SD). Clarify if PO₂ is arterial (ABG) PaO₂ not SpO₂.

2. Statistical reporting: Use consistent formatting for p-values and CIs (e.g., p = 0.017, HR 0.86 [95% CI 0.78–0.95]). Avoid phrases like “P-overall” without explanation - define tests used.

3. Subgroup analyses: State that subgroup analyses are exploratory, adjust for multiplicity (or state they are unadjusted), and present N in each subgroup. The stroke interaction (p = 0.028) needs cautious language and possible biological hypothesis.

4. Ethics / data access: You appropriately state MIMIC-IV access; also add a statement on data/code availability and R packages and versions used (you cite R 4.4.1 - also list packages e.g., survival, rms, Boruta, SHAP).

5. Limitations: Add explicit limitations section (single-center design proxy via MIMIC, residual confounding, measurement timing, no causal inference, need for external validation, changes in ARDS care across 2008–2022).

6. Causal claims: Avoid language implying that the ratio “guides” ventilator use or “improves outcomes” without prospective interventional evidence. Rephrase to “may help risk-stratify and identify patients who warrant closer monitoring; prospective validation is required before clinical implementation.”

**Do you want your identity to be public for this peer review?** For information about this choice, including consent withdrawal, please see our Privacy Policy

Reviewer #1: No

Reviewer #2: No

---

## [Author Response · Author response to Decision Letter 1]

21 Oct 2025

Dear Academic Editor and Reviewers,

We sincerely thank you for your thoughtful and constructive comments, which have greatly improved the quality of our manuscript.

We have carefully addressed all comments and revised the manuscript accordingly. A detailed, point-by-point response is provided in the uploaded file titled “Response to Reviewers.docx.”

We hope that the revised version satisfactorily addresses all concerns and will be suitable for publication in PLOS ONE.

Sincerely,

The Authors

---

## [Editor Report · Decision Letter 1]

26 Oct 2025

Dear Dr. deng,

Thank you for submitting your manuscript to PLOS ONE. After careful consideration, we feel that it has merit but does not fully meet PLOS ONE’s publication criteria as it currently stands. Therefore, we invite you to submit a revised version of the manuscript that addresses the points raised during the review process.

**ACADEMIC EDITOR: Minor revision**

We look forward to receiving your revised manuscript.

Kind regards,

Marwan Salih Al-Nimer, MD, PhD

Academic Editor

PLOS ONE

Journal Requirements:

Additional Editor Comments:

Take a look to the order of the authors. In the revised manuscript is differed from that in the first submission. It requires correction

---

## [Author Response · Author response to Decision Letter 2]

28 Oct 2025

Dear Editorial Office,

As requested, we have uploaded the completed Authorship Change Request Form (including all author signatures) to correct the author order in our manuscript (PONE-D-25-24350R2).

All authors have reviewed and approved the correction.

Thank you very much for your kind guidance.

Best regards,

Jiang-Tao Deng

---

## [Editor Report · Decision Letter 2]

7 Dec 2025

Association of the PaO₂/RDW Ratio With 7-Day Mortality and Risk of Early Invasive Mechanical Ventilation in ICU Patients With Delirium Associated With ARDS: A Retrospective Cohort Study From the MIMIC-IV Database

PONE-D-25-24350R2

Dear Dr. lop - Deng,

We’re pleased to inform you that your manuscript has been judged scientifically suitable for publication and will be formally accepted for publication once it meets all outstanding technical requirements.

Kind regards,

Marwan Salih Al-Nimer, MD, PhD

Academic Editor

PLOS One
---

## [Editor Report · Acceptance letter]

PONE-D-25-24350R2

PLOS One

Dear Dr. Deng,

I'm pleased to inform you that your manuscript has been deemed suitable for publication in PLOS One. Congratulations! Your manuscript is now being handed over to our production team.

Kind regards,

on behalf of

Professor Marwan Salih Al-Nimer

Academic Editor

PLOS One